# An Automatic HEp-2 Specimen Analysis System Based on an Active Contours Model and an SVM Classification

**Donato Cascio *** , **Vincenzo Taormina and Giuseppe Raso**

Department of Physics and Chemistry, University of Palermo, 90128 Palermo, Italy;
taormina.maltese@gmail.com (V.T.); giuseppe.raso@unipa.it (G.R.)
* Correspondence: donato.cascio@unipa.it; Tel.: +39-091-238-99050



**Featured Application: In this paper we describe a complete system (fluorescence intensity classification, image preprocessing, cell segmentation, cell classification, and image staining patterns classification) to support the autoimmune diagnostics in HEp-2 image analysis. The system has been tested on a heterogeneous public database; heterogeneity comes from the geographical origins of indirect immunofluorescence (IIF) images and acquisition instrumentation adopted. This system is able to recognize six different patterns—homogenous, speckled, nucleolar, centromere, nuclear dots, and nuclear membrane.**

**Abstract:** The antinuclear antibody (ANA) test is widely used for screening, diagnosing, and monitoring of autoimmune diseases. The most common methods to determine ANA are indirect immunofluorescence (IIF), performed by human epithelial type 2 (HEp-2) cells, as substrate antigen. The evaluation of ANA consist an analysis of fluorescence intensity and staining patterns. This paper presents a complete and fully automatic system able to characterize IIF images. The fluorescence intensity classification was obtained by performing an image preprocessing phase and implementing a Support Vector Machines (SVM) classifier. The cells identification problem has been addressed by developing a flexible segmentation methods, based on the Hough transform for ellipses, and on an active contours model. In order to classify the HEp-2 cells, six SVM and one *k*-nearest neighbors (KNN)classifiers were developed. The system was tested on a public database consisting of 2080 IIF images. Unlike almost all work presented on this topic, the proposed system automatically addresses all phases of the HEp-2 image analysis process. All results have been evaluated by comparing them with some of the most representative state-of-the-art work, demonstrating the goodness of the system in the characterization of HEp-2 images.

**Keywords:** IIF images; Hough transform; active contours model; cell segmentation; SVM; KNN; ROC curve

## 1. Introduction

Autoimmunity is the phenomenon for which the immune system activates its mechanisms towards molecules, cells, and structures of the same organism to which it belongs. The diseases caused by this phenomenon, which are defined as "autoimmune", are becoming increasingly widespread and include some of the most serious and penalizing conditions for the quality of life of those affected [1]. The diagnosis of autoimmune diseases is based on the finding of auto-antibodies, that is, antibodies directed against components of the same organism that produced them. Anti-nucleus antibodies (ANA) are a group of antibodies produced by the immune system that can mistakenly recognize the structures of the organism they belong to (autoantibodies). The ANA test identifies the presence of these autoantibodies in the blood. ANAs are the most recognized markers for

the identification of an autoimmune process and, therefore, allow the exclusion of the presence of pathologies characterized by similar signs and symptoms. Positivity to the ANA test, tested by means of the Indirect ImmunoFluorescence (IIF) method, and performed by analyzing patterns and fluorescence intensity [2], is associated with multiple autoimmune diseases. The IIF test is particularly sensitive, but poorly specific. This is because ANA antibody titers tend to present themselves superior to the norm, in various non-pathological conditions. Furthermore, fluorescence intensity and fluorescence pattern analysis is particularly difficult, due to the similarity between different classes, and in any case is linked to the operator's experience [3]. The fluorescence pattern observed on the microscope (homogenous, speckled, nucleolar, cytoplasmic, centromere, nuclear dots, etc.) is specific, according to the nature of the self-antigen and its location in the cell. For these reasons, two senior immunologists (for double-reading) with strong experience in fluorescent image interpretation, are quite often needed. However this condition is not respected in all immunology laboratories involved in the diagnosis. The introduction of new modern approaches, based on computer systems, is an economic and effective support for the diagnosis of autoimmune diseases [4,5]. Computer Aided Diagnosis (CAD) systems have been widely proposed in different areas of medicine and with different objectives, such as second-reading, improving the speed of the diagnostic processes, training physicians for special tasks, etc. [6,7]. In recent years, the diagnostic support of CAD systems has been proposed for an automatic human epithelial type 2 (HEp-2) images classification. In this paper we describe a complete system (fluorescence intensity classification, image preprocessing, cell segmentation, cell classification, and image staining patterns classification), to support the autoimmune diagnostics in a HEp-2 image analysis. The system has been tested on a heterogeneous public database; heterogeneity comes from the geographical origins of the IIF images and the acquisition instrumentation adopted. This system is able to recognize six different patterns—homogenous, speckled, nucleolar, centromere, nuclear dots, and nuclear membrane. Figure 1 shows examples of each class.

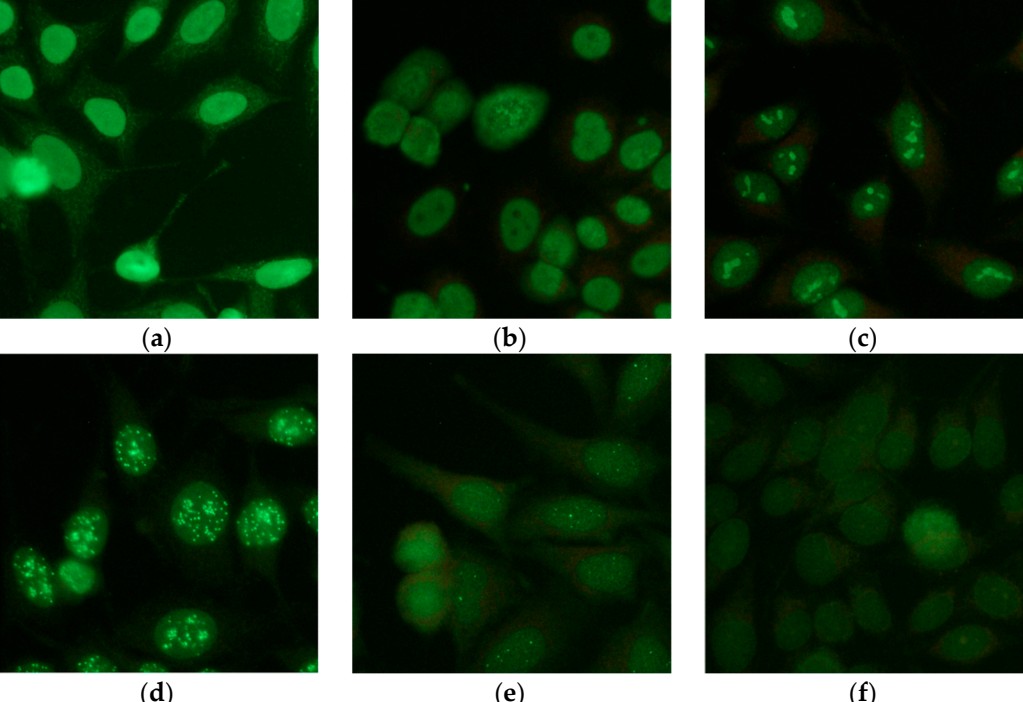

**Figure 1.** Examples of staining patterns in the IIF images; (**a**) homogenous; (**b**) speckled; (**c**) nucleolar; (**d**) centromere; (**e**) nuclear dots; (**f**) and nuclear membrane.

## 1.1. Related Work

This section briefly reviews previous work related to the HEp-2 image analysis, in the context of ANA testing. In past years, the scientific community has shown a great interest in the analysis of IIF images, and in particular for the development of CAD systems. Various aspects of the entire analysis flow have been addressed by different research groups [8]. As a consequence of this, in the literature, it is possible to find work concerning the image acquisition [9], image preprocessing [10], and cell segmentation [11,12]. Another group of papers (as a consequence of the various competitions that were organized on the subject [13–15]), relate directly to the HEp-2 cells classification or staining pattern recognition. However, for studies belonging to this last group, the segmentation phase of the individual cells was performed, manually, and the fluorescence intensity analysis has not been addressed. The I3A (Indirect immunofluorescence images Analysis) contest [15] was the most recent in this series of contests. In this competition, Nanni et al. [16] proposed an approach for the automatic image classification of HEp-2 cells, using several texture descriptors and based on a set of Support Vector Machines (SVMs) combined by the sum rule. The accuracy obtained on the test set was 79.85%.

Ensafi et al. [17] proposed a HEp-2 cell classification technique, based on the superpixel approach and by using a sparse coding scheme. In particular, the authors used superpixels to obtain the image patches which contain more 'informative' features. This method obtained the best accuracy, among other methods, on the ICPR2012 dataset—for the cell-level classification, an accuracy of 79% was obtained.

In our previous studies [18,19] we have tackled the problem of fluorescent pattern classification, we proposed a classification approach based on a one-against-one (OAO) scheme. To do this, a large number of classifiers have been trained. The classification processes have been differentiated; for each classification a different process (for procedures and parameters) was developed, aimed at the extraction of the most suitable feature, for each pattern to be searched.

Several work in the literature address the individual steps of the work-flow, nonetheless, integrating such steps and assessing their effectiveness, as a whole, is still an open challenge. In fact, probably because of the difficulty of realization or because public databases have only been created in recent years, there have been few studies in which the realization of a complete automatic CAD system for the support of the autoimmune disease diagnosis in IIF images has been attempted, and consequently, there are very few studies where performance comparisons have been made [20,21].

Di Cataldo et al. [22] presented a modular method (called ANAlyte) which is able to automatically characterize IIF images in terms of fluorescence intensity and fluorescent pattern. Performance analysis was performed using images from two public databases, for a total of 99 images used. The performances obtained for the fluorescence intensity and fluorescent pattern were 85% and 90%, respectively.

Chung-Chuan Cheng et al. [23] have developed a system able to detect the cell patterns of IIF images. This system segments and classifies cells, but does not analyze the fluorescence intensity of the image. The performance analysis, conducted on only 17 patients, showed an average accuracy of 96.9%. The time needed for the whole procedure was approximately 30 min.

Elgaaied Benammar et al. [24] have optimized and tested a CAD system on HEp-2 images which is able to recognize seven fluorescence patterns. The system searches and classifies positive and negative mitosis, within the image; the classification of mitosis occurs by using two neural network classifiers, the final decision-making process for the detection of fluorescence pattern is achieved by using a K-Nearest Neighbors classifier. The results showed 85.5% of intensity fluorescence accuracy and 79.3% of pattern fluorescence accuracy.

## 1.2. Our Contributions

In this paper, a CAD system that is able to identify the fluorescence intensity and the fluorescence patterns in IIF images, has been presented. In particular, the analysis of fluorescence intensity, for the positive/negative detection, is carried out using an SVM classifier. A segmentation phase is included, which is capable of isolating the cells related to the different types of investigated patterns, using

an initialization method based on randomized Hough transform for ellipses, and an active contours model algorithm (ACM). The problem of staining the patterns classification, in this work, has been addressed by developing a differentiated analysis method for each pattern. In many multiclass problems, it is more efficient to use a binary approach, implementing a classifier for each class for the discrimination process [25]. In the work presented here, in addition to a differentiated classification step, the preprocessing and features extraction steps are also differentiated. The system is able to recognize the following staining pattern—homogenous, speckled, nucleolar, centromere, nuclear dots, and nuclear membrane. Starting from a set of preprocessing functions, all their possible couplings, in terms of class accuracy, were evaluated, and the best performing combination for each pattern was chosen. For each class under analysis, a significant number of features (108) was extracted, and a feature reduction phase, based on linear discriminant analysis (LDA), was performed with the aim of selecting the characteristics that are able to characterize the specific staining pattern. A classification approach based on the one-against-all (OAA) scheme has been implemented; six SVMs have been implemented to classify the IIF images. For a robust cell-pattern association, a final k-nearest neighbors (KNN) classifier, with the aim of synthesizing the six SVM results and obtaining a robust classification, has been implemented.

## 2. Materials and Methods

### 2.1. Database

The development of a CAD system is intimately linked to a data collection. In this work the dataset provided by the AIDA (AutoImmunité, Diagnostic Assisté par ordinateur) project [24] was used. In this project, using a uniform approach, seven immunology services (three Tunisian and four Sicilian) contributed to collect images of the IIF test on HEp-2 cells. These images corresponded to the routine IIF technique performed in different hospitals for an autoimmune disease diagnosis and were, thus, reported by senior immunologists. Each image and related report was stored in a common database created in the project. Manufacturers of kits and instruments employed for the ANA testing were different site-to-site; the following automated systems solution for the processing of Indirect Immunofluorescence tests have been used—IF Sprinter from Euroimmun, NOVA from INOVA diagnostic, and Helios from Aesku. They were used to carry out the serial dilutions and a dilution of 1/80was considered to be positive. The images had 24 bits color-depth and were stored in the common image file formats. The database acquired during the AIDA project consisted of two parts—one part was public, the other was private.

The public "AIDA_HEp-2" Database was a subset of the full AIDA database, where three physician experts(independently) have expressed a unanimous opinion when reporting. This is available to the scientific community and, to our knowledge, it is the largest HEp-2 images public database that is most representative of real cases, including a variety of single and multiple patterns; 2080 images for which there is a triple concordance of reports.

The database contains fluorescence positive sera with a variety of more-than-twenty staining patterns. In each image a single or multiple pattern can be present. The patterns terminology is in accordance with the "International Consensus on ANA Patterns" (ICAP): "www.anapatterns.org" [26].

The public part of the AIDA Database consists of 2080 images, composed as follows: 998 patients (261 males, 737 females), 1498 images show positive fluorescence intensity, 582 show negative, 479 refer to males and 1601 to females. Among the images with positive fluorescence, those relating to patterns belonging to the six typologies object of the present research are a total of 220, their distribution is shown in Figure 2. The public database can be downloaded, after registration, from the download section of the AIDA project site (http://www.aidaproject.net/downloads).

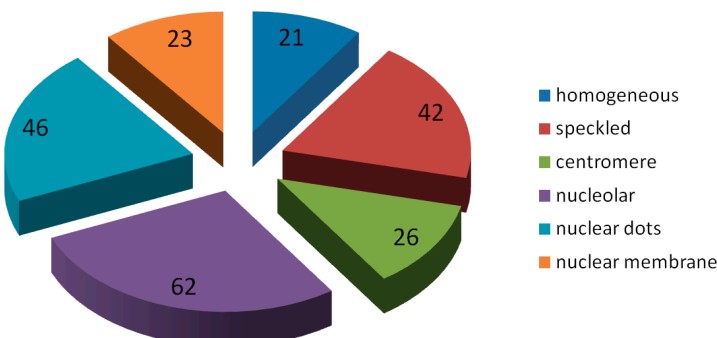

**Figure 2.** Staining patterns distribution of the test set.

The private part of the AIDA Database is structured in the same way as the public part, and it has about 20,000 positive and negative images. However, among all these images only about 3000 have triple concordance of reports. This part of the AIDA database is only accessible to the partners who participated in the project.

The system proposed here has been trained and optimized on the private part and tested on the public part of the AIDA database. In the training–tuning phase, in order to make the best use of the data, the leave-one-specimen-out cross-validation technique have been used. This method consists of leaving out one specimen, rather than leaving out a single image (or a single cell) for the construction of the training set; images of the same specimen, belonging to the same patient, are similar (in terms of the average intensity and contrast) and introduce bias. The specimen left out is used for validation. Regarding the analysis of fluorescence intensity, as in this case, the wells at our disposal were statistically much greater; during training–tuning we proceeded with a leave-ten-specimens-out (i.e., leaving out ten specimen).

### 2.2. CAD Workflow

The automatic system presented here is able to identify, on the IIF images, the fluorescence intensity and fluorescence pattern. In particular, the analysis of fluorescence images for the positive/negative detection is carried out using an SVM classifier. The images that are fluorescent positive are analyzed for staining pattern classification. The positive image is then segmented and the contained cells are extracted. At this point, every single cell extracted is analyzed by six processes and at the end is associated with one of the possible patterns object of the present research. Figure 3 shows the CAD working flow. The system is able to recognize the following staining patterns—homogenous, speckled (fine and coarse), nucleolar, centromere, nuclear dots, and nuclear membrane. The CAD system presented here adopts a non-standard pipeline for a supervised image classification. Indeed, in several multiclass classification problems, it is preferable to use different binary classifiers for the classification phase of each class (usually in numbers equal to the classes to be analyzed. In this system, in addition to differentiating the classification stage, by implementing six SVMs classifiers, the preprocessing, and feature extraction steps are differentiated as well. In the multiclass classification problems (specifically with the OAA scheme) the outputs of the binary n-classifiers are evaluated; usually the classifier that produced the maximum output value is identified, and the final association of the generic Region of Interest (ROI) is assigned to the relative class. This procedure may not be very robust, for example it could happen that one of the classifiers produces output values on average higher than the other classifiers, thus, completing the final classification. In this work it was decided to evaluate the outputs produced by means of a further classifier. The classifier KNN has been chosen, as it allows a simple multi-class implementation; this classifier, using examples belonging to the classes to be analyzed, associates the generic element with the class having the most examples close to it.

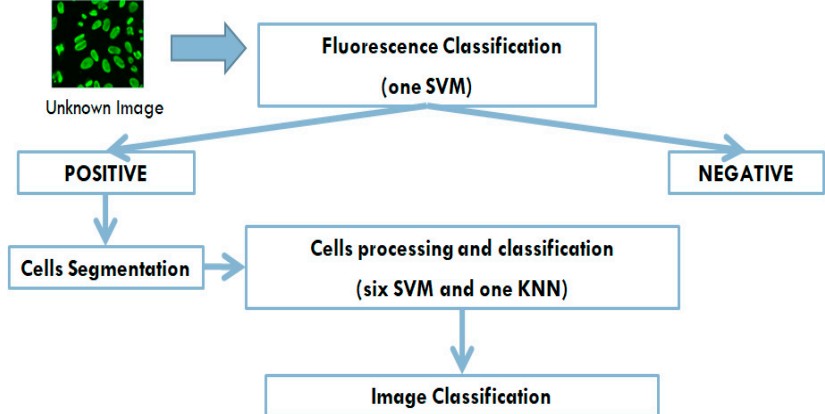

**Figure 3.** The Computer Aided Diagnosis (CAD) working flow—the system aims to reproduce the operations flow made by the immunologist, by making a classification of fluorescence only for not-negative images, and by operating a cells classification which will then use the results of these classifications, to provide a final output.

### 2.3. Fluorescence Intensity Classification

As it is aimed at identifying a patient's positivity/negativity to the test, the fluorescence intensity classification is an important phase. Moreover, with regards to the CAD system, it will be the result of this phase that will establish (in the case of a positive output) if the execution of the analysis steps—aimed at identifying the staining patterns present in the image—will be carried out.

This phase was addressed by analyzing the generic image in its entirety; a set of features are extracted from the entire image and these are used by an SVM classifier, in order to associate the generic image with the positive/negative classes. SVM has been widely used in biomedical research [27–32] and this is certainly linked to the results of the classification, but also to the advantage that this type of classifier depends on a few parameters.

The image is initially subjected to a preprocessing phase. An intensive analysis of the preprocessing function combinations was conducted, aimed at maximizing the performance. In particular, this analysis led to the selection of the following preprocessing operations, performed in the order indicated:

(1) selection of the green channel;
(2) stretching;
(3) median filter (kernel $9 \times 9$);
(4) maximum entropy threshold;
(5) filling;
(6) remove cells on boundary.

A considerable number (about 150) of characteristics intensity-based and texture-based, were extracted from the preprocessed image. To achieve a reduction in complexity and an appropriate selection of features, the LDA method was used. In this way only twelve features were selected for the fluorescent intensity classification. An SVM classifier, with a Gaussian RBF (Radial Basic Function) kernel with twelve inputs, was then implemented. The classifier parameters were obtained with the "grid-search" method (as described in Section 3.1).

### 2.4. Cell Segmentation

This module performs the automated segmentation of the individual HEp-2 cells. Such a task is one of the most challenging of the automated IIF analysis, because the segmentation algorithm has to cope with a large heterogeneity of shapes and textures. Moreover, very often, inside the HEp-2 images the cells are partially overlapped with a consequent separation problem.

In this work, the segmentation phase was carried out by developing the following three steps:

(1)　Pre-segmentation: Aimed at identifying regions of interest ROIs (Regions of Interest).
(2)　Randomized Hough transform for ellipse: Aimed at identifying the ellipse that best characterizes the generic cell.
(3)　Active contours model: Starting from an elliptic curve, evolve expanding towards the cellular contour.

### 2.4.1. Pre-Segmentation

This first step has the task of identifying the ROIs, to be further analyzed in the subsequent steps of the method. In this step the image is preprocessed and a segmentation by means of a threshold algorithm is implemented. Since the cells in the image are very often overlapping, the threshold methods, or the region growing methods, do not always allow the correct separation of the same; in this case, one of the possible choices, in the development of an automatic system, would be to discard the ROI from the next analysis and, as a result, losing in the segmentation accuracy.

The pre–segmentation method, thus obtained, is composed of the following operations, performed in the order indicated below

(1)　selection of the green channel;
(2)　anisotropic filter (Ofeli library: http://www.ofeli.org/download);
(3)　adaptive thresholding (OpenCV library: adaptive thresholding (OpenCV library: https://docs.opencv.org/3.4/d7/dd0/tutorial_js_thresholding.html));
(4)　removal of small ROIs.

### 2.4.2. Randomized Hough Transform for Ellipses

For the realization of an accurate segmentation, able to identify the different types of patterns and to separate any overlapping cells, in this work, a geometrical method was applied for the initialization of the final step, based on an active contour model. In fact, the cells can be approximated to ellipses and then searched for geometrical figures; the identification of the equivalent ellipse allows the definition of the center and dimensions for the generic cell. Many methods have been developed for the identification of geometric figures, certainly among the most performing is the Hough transform.

Ellipse detection is an important class of computer vision algorithms because an ellipse is the 2D projection of a circle in an image, from a 3D scene. Circular and elliptical shapes are often present in both nature and constructed environments, such as those found in a factory or a city, and computer vision systems can use information about the circles or ellipses in a given scene, to perform some useful task. For this reason, many methods have been developed for the identification of elliptical objects in images [33–35].

Due to the many parameters involved in the detection of ellipses, the various methods developed often limit their recognition dimensions, or search for sub-images. A complete search would be computationally not feasible. The traditional approach for ellipse detection using the Hough technique is similar to a line or circle detection. This approach is not only memory expensive but also computationally intensive. In this work, for an ellipse detection, a randomized Hough Transform with result clustering is used [36]; the method uses only a one dimensional accumulator for ellipse voting, and reducing algorithm complexity. In order to be processed by the actual ellipse detector, a preprocessing phase, composed of the following steps, is used:

(1)　noise reduction;
(2)　grey-scale transform;
(3)　edge detection;
(4)　 binarization.

In order to reduce the calculation time and memory usage, in this work, the randomized Hough transform was applied to the sub-images containing the ROIs obtained from the pre-segmentation. Figure 4 shows an example of the result of the transformation applied on a sub-image containing

two contiguous cells. As can be seen from the Figure, the method identifies the two ellipses that best characterize the two contained cells.

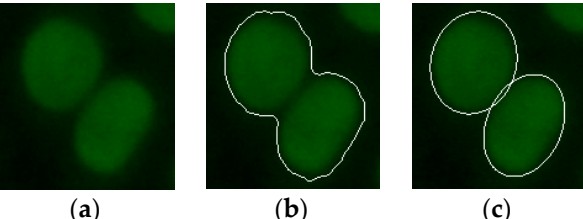

| (**a**) | (**b**) | (**c**) |

**Figure 4.** (**a**) An example of an indirect immunofluorescence (IIF) sub-image containing two attached fluorescent cells; (**b**) the pre-segmented IIF sub-image; and (**c**) the relative result obtained from the randomized Hough transform method.

### 2.4.3. Active Contours Model

The problem of the identification of the contours for patterns that are visually very different and need to be separated from overlapping cells inside the Hep-2 image, has been addressed in this work, by implementing a method of active contours. This family of algorithms has the advantage that, if properly initialized, they can converge on the correct contour of objects and since the evolution involves a non-punctual analysis of the image, they allow the separation of contiguous but visually distinct objects. The characteristics of the active contour model algorithms allow to address the problem of cells overlapping. In particular, the method used is a model of active contours, based on the techniques of curve evolution, Mumford–Shah functional for segmentation and level sets [37], which is a Matlab 2017 (MathWorks, Natick, Massachusetts, USA) function. This model can detect objects whose boundaries are not necessarily defined by a gradient. The method involves minimizing an energetic functional which can be seen as a particular case of the minimal partition problem. The stopping term does not depend on the gradient of the image, as in the classical active contour models, but is instead related to a particular segmentation of the image. The method is particularly stable and allows a good convergence, even if the initial conditions are not optimal. Additionally, this model does not require image smoothing, even if it is very noisy, and in this way the locations of boundaries are very well-detected and preserved.

The result of the Hough transform was used to initialize the active contour. In particular, the ellipse identified on the generic cell has been reduced in size, halving the axle shafts, in order to obtain an evolution of contour, which in expansion tend to be the desired cell boundary. Figure 5 shows an example of the initialization and evolution of an active contour obtained on a pattern, which is particularly difficult to segment—the cytoplasmic pattern. The average iterations number for convergence of the proposed method is around 80.

Figure 6 shows an example of segmentation, which shows the lack of accuracy obtained from the pre-segmentation (the nucleoli contained within the cell have been identified by committing over-segmentation) and the correct identification of the contours obtained from the ACM method.

In this work, the Dice index [38] has been chosen as a measure of the goodness of segmentation. The Dice index, also called the overlap index, is the most used metric in validating image segmentations [39], this index was used for a direct comparison between the ground truth and automatic segmentations. More generally, the Dice measures the spatial overlap between two segmentations, A and B target regions, and is defined as:

$$Dice(A,B) = \frac{2(A \cap B)}{(A \cup B) + (A \cap B)} \tag{1}$$

where the symbol $\cap$ represents the intersection, and the symbol $\cup$ is the union.

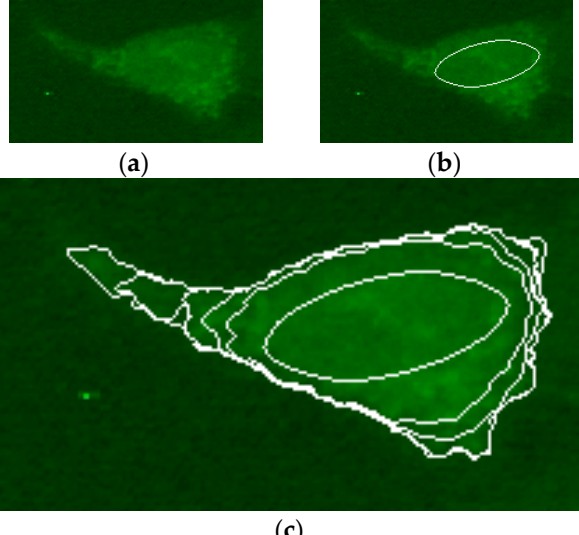

**Figure 5.** (**a**) An example of IIF sub-image containing cytoplasmic pattern; (**b**) the initial ellipse for the process of active contours; and (**c**) the contour evolution (with a step of 25 iterations) from the initial condition to the identification of the pattern.

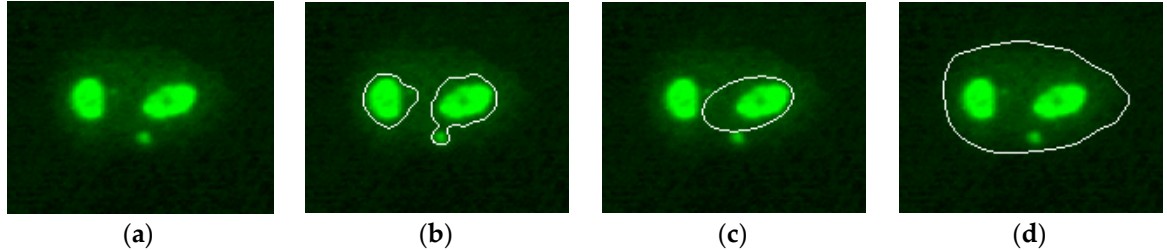

**Figure 6.** (**a**) An example of IIF sub-image containing nucleolar pattern; (**b**) pre-segmented sub-images; (**c**) relative result obtained from the randomized Hough transform method; and (**d**) the result of the active contours model (ACM)) algorithm.

The ground truth segmentation was obtained with a manual segmentation performed by an expert in the field. A total of 95 images were manually segmented, their distribution is shown below:

- 15 homogenous;
- 16 speckled;
- 19 nucleolar;
- 15 centromere;
- 23 nuclear dots;
- 7 nuclear membrane.

## 2.5. Preprocessing for Pattern Classification

Figure 7 shows the flow of operations, adopted in this work, for the cellular classification—the generic segmented cell is simultaneously processed by six processes, obtaining six separate outputs representing how the cell resembles each of the six staining patterns analyzed in this work. The choice of methods, features, and parameters was performed automatically, using Accuracy as a figure of merit. The main benefit of this pipeline consists of offering of a good explanatory faculty, based on an easily explainable principle. The image classification is achieved by means of the cell classification. The rate of presence of the individual patterns, within the image, is evaluated and the generic image is associated with the pattern that has a higher rate.

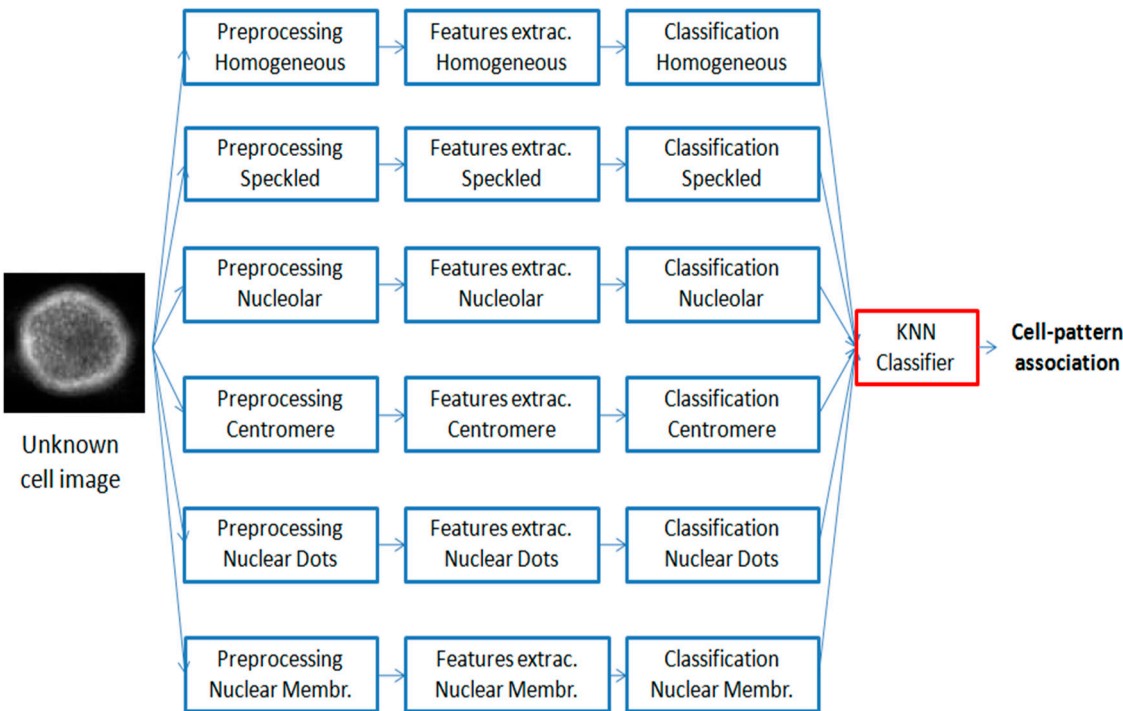

**Figure 7.** Pipeline of the cell classification method—the generic Region of Interest (ROI) is simultaneously analyzed by six processes, thus, obtaining six separate outputs showing how the cell resembles each of the six classes analyzed in this work; the six classification results are used as inputs of a *k*-nearest neighbors (KNN) classifier, for a robust cell-pattern association.

In a pattern recognition process, in particular when, as in this case, the objects to be discriminated are very heterogeneous, the image preprocessing has considerable importance, since from this phase the features extraction will be strictly dependent. For this reason, in this work, the preprocessing phase has been differentiated for each type of pattern to be recognized. Starting from a set of preprocessing functions, all their possible couplings, in terms of class accuracy, were evaluated, and the best performing combination for each pattern was chosen. Table 1 shows the preprocessing functions used.

**Table 1.** Preprocessing functions analyzed.

| Abbreviation | Function Description | Reference |
|---|---|---|
| Nt | Nothing | - |
| Fs | FAS (Filter Alternate Sequential) | http://www.pkuwwt.tk/ofeli/doc/index.html |
| Eq | Equalization | http://docs.opencv.org |
| Ch | CLAHE (Contrast Limited Adaptive Histogram Equalization) | http://docs.opencv.org |
| Md | Median filter | http://docs.opencv.org |
| Gs | Gaussian filter | http://docs.opencv.org |
| Ge | Gaussian-Enhancement: | http://docs.opencv.org |
| Di | Dilatation | http://docs.opencv.org |
| Er | Erosion | http://docs.opencv.org |
| Ng | Nagao | http://www.pkuwwt.tk/ofeli/doc/index.html |
| Bl | Bilateral | http://docs.opencv.org |
| Op | Opening | http://docs.opencv.org |
| Cn | Contrast normalization | - |

Many of the functions in the table have been analyzed using several parameters (e.g., median kernel of size $3 \times 3$, $5 \times 5$, and $7 \times 7$ have been analyzed); overall a total of thirty-seven functions were analyzed. The analysis carried out allowed the determination of the best configuration for each analyzed pattern, the result is shown in Table 2.

**Table 2.** The best preprocessing for each pattern.

| First Function | Second Function | Classification Category |
|---|---|---|
| Er | Ch | homogenous |
| Di | Cn | speckled |
| Er | Ch | Nucleolar |
| Di | Cn | Centromere |
| Fs | Nt | nuclear dots |
| Fs | Nt | nucleolar membrane |

## 2.6. Features Extraction

The differences between staining patterns in the IIF image are due to the type and location of the fluorescence. Then, the identification of the type pattern can be understood as the analysis of the presence and distribution of bright/dark structures in the IIF image. In order to characterize IIF images, it is, therefore, natural to transform visual information into the characteristic quantities that are capable of describing the presence and distribution of bright/dark regions within cells. The features can be divided into three families [40–43]:

(1) intensity features;
(2) shape features; and
(3) texture features.

Since the fluorescence patterns show remarkably different characteristics, in order to transform the visual content of all of them into suitable features, a considerable number of features have been extracted from the ROIs and a selection of the features is made for each pattern. Feature selection is the process of selecting an optimum subset of features from a set of potentially available features in a given problem domain.

Specifically, four different quantization intensity levels were analyzed. The quantizations explored were: 256, 128, 64, and 32 gray levels. The intensity quantization, as is known, affects the quality of the representation of the image. Reducing the number of bits to represent intensity, compresses the storage space, but causes the image quality to deteriorate. In this work, the different quantizations aim to highlight the different aspects of ROI; those with more bits have more details, while those with fewer bits show the shapes more clearly.

The following set of rotational invariant features has been extracted at each quantization level:

- Intensity features (6 features): Mean value, standard deviation, ratio of the standard deviation to the mean value, entropy, skewness, and kurtosis [44];
- Shape features (12 features): Area, perimeter, convex area, mean radius, standard deviation of radius, ratio of the standard deviation to the mean value, maximum radius, anisotropy, entropy of the contours gradient, fractal index, eccentricity, and circularity [45];
- Texture features (9 features): Contrast, convex deficiency, roundness, compactness, solidity, inertia of co-occurrence matrix, entropy of histogram of oriented gradients (HOG), entropy of histogram of amplitude gradients (HAG), and Euler's number [46,47];

Overall, each ROI has been described by twenty-seven characteristics obtained at four different quantization levels, for a total of one hundred and eight features.

## 2.7. Features Selection Based on LDA

Feature selection strategies were applied in order to select a limited set of optimal features able to improve the accuracy of the staining pattern classifier. The main idea of feature subset selection is to remove redundant or irrelevant features from the dataset, as they can negatively influence the classification accuracy and lead to an unnecessary increase of computational cost.

There are many methods proposed in the literature for the decrease in dimensionality, used in supervised or unsupervised classification problems, such as sequential forward search [31,48], random forest algorithm [49,50], and ANOVA feature selection [51].

Discriminant analysis algorithms have been used for dimensionality reduction and feature extraction in many applications of computer vision [52,53]. In most discriminant analysis algorithms, the transformation matrix is found by maximizing the Fisher-Rao's criterion [54].

Linear Discriminant Analysis (LDA) is probably the most well-known discriminant analysis technique. This method assumes that the *C* classes to which the data belong, are homoscedastic, that is, their underlying distributions are Gaussian with common variance and different means. The LDA method provides the (*C*-1)-dimensional subspace that maximizes the between-class variance and minimizes the within-class variance, in any particular data set. In other words, it guarantees maximal class separability and, possibly, optimizes the accuracy in later classifications. In this work, the LDA method was used. For each classifier, the set of features with the highest classifying power has been identified. The LDA method allowed a reduction of complexity, on average, of about a factor of eight.

## 2.8. Classification

The classification in this multiclass analysis work represents the most complex task, as it requires a structured and computationally expensive training process. It is known that to address a problem of classification on the M-classes, it is effective to break down the problem into an appropriate number of two class problems. In this regard, the most used strategies are: One-Against-One (OAO) and One-Against-All (OAA) [55]. The OAO approach involves constructing a classifier for each pair of classes, resulting in an M(M − 1)/2 classifier; each classification gives one vote to the winning class and the sample is labeled with the class having the most votes. The OAA approach is the most commonly used method and involves the decomposition of the classification into M-classes, in M two classes classifications. The choice on how to evaluate the overall results of the M classifiers often leads to label the sample with the class that has obtained the highest classification result.

In this work, an OAA approach was used. For each class pattern, one classifier is implemented (e.g., homogenous vs. all), overall they have been implemented as six classifiers. The evaluation of the outputs of the six SVM classifiers has been solved by implementing a KNN classifier, with the aim of synthesizing the six results and obtaining a robust classification. There are also some successful implementation of the SVM- and KNN-based CAD system in medical applications, which shows a good reason for the potential use of this machine learning method for cell classification [56].

In this work, the Gaussian kernel was adopted, which is an example of a radial basis function (RBF). Gaussian kernel can be represented as:

$$k(x, x_i) = e^{-\gamma \|x - x_i\|^2} \quad \gamma > 0 \tag{2}$$

The linear kernel is a special case of RBF [57,58], since the linear kernel with a penalty parameter C has the same performance as the RBF kernel with some parameters (C, $\gamma$).

## 2.9. Functions and Parameters Selection

In order to identify the optimal functions and parameters for the differentiated analysis of the searched patterns, in this work, an iterative method of configuration analysis was implemented, aimed at maximizing the class accuracy. In Figure 8, the flow chart of the iterative method used for each of the six binary classification is shown.

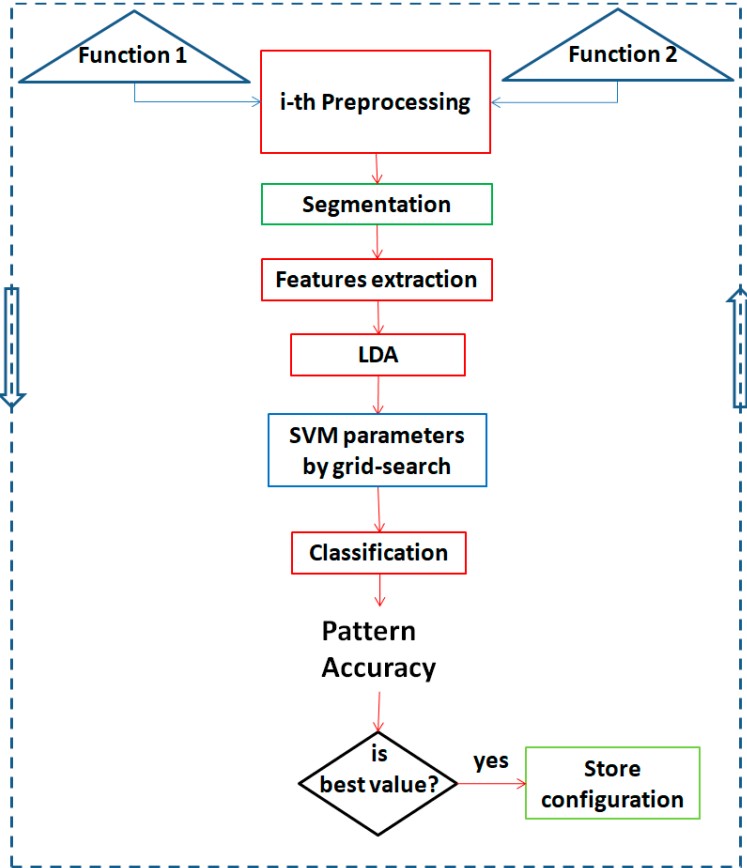

**Figure 8.** Flow chart of the iterative method used for the optimization of each developed binary classification. Two preprocessing functions are extracted from among those present in Table 1, the flow of operations envisaged by the system is performed (preprocessing, segmentation, features extraction, linear discriminant analysis (LDA), Support Vector Machines (SVM) best parameters, classification, and accuracy calculation). If the obtained accuracy result is the best, the configuration is stored (in terms of preprocessing functions used, selected features and SVM parameters obtained). In any case, the process is cycled for all configurations of the two functions that were obtainable, starting from Table 1.

This method allows the identification, for each binary classification, of—two preprocessing functions, a set of best features, and two parameters (C and $\gamma$) for the SVM classifier. As can be seen from Figure 8, all pairs of preprocessing functions among those implemented (shown in Table 1) are analyzed and the relative preprocessing is performed. The images, thus obtained, after the segmentation step, are used for the extraction of features. The next phase has the aim to reduce the dimensionality of the features, according to the LDA method. For each binary classification, the configuration of the preprocessing functions, features, and classifier parameters that have the maximum accuracy, is stored. Finally, the optimization method leads to a parallel process system for patterns analysis, as represented in Figure 7.

## 3. Results and Discussion

### 3.1. Selection of the SVMs Parameters

The two parameters in the RBF kernel, C and $\gamma$, were analyzed, in order to identify the optimal configuration for each implemented SVM classifier. To this end, a method of "searching the grid" on the C and $\gamma$, using leave-one-specimen-out was implemented [59].

For each binary classifier, various pairs of the C and $\gamma$ values were tried and the one with the best accuracy was taken. A practical method to identify good parameters makes use of the exponentially growing sequences.

In this work, the analyzed values for the C and $\gamma$ were:

$$C=2^{-5}, 2^{-4}, \ldots, 2^{10} \quad \gamma = 2^{-10}, 2^{-9}, \ldots, 2^{2}$$

For each OAA classifier implemented, the analyzed grid had sizes equal to $16 \times 13$, for a total of 208 grid-points.

With regard to the intensity analysis and the relative SVM classifier used, the analysis of the parameters were carried out to achieve the following values:

$$C = 64 \quad \gamma = 0.25$$

Table 3 reports the best values of the parameters C and $\gamma$, related to the classifiers implemented for the identification of fluorescence patterns and obtained for each class pattern.

**Table 3.** The C and $\gamma$ parameters obtained for each one-against-all(OAA) classifier.

| Class | C | $\gamma$ |
|---|---|---|
| homogenous | 256.0 | 0.125 |
| speckled | 32.0 | 0.25 |
| centromere | 256.0 | 0.25 |
| Nucleolar | 128.0 | 0.50 |
| nuclear dots | 256.0 | 0.0156 |
| nuclear membrane | 128.0 | 0.0625 |

### 3.2. Fluorescence Intensity Results

The fluorescence intensity classification method (described in Section 2.4) was analyzed using all images in the public AIDA database (2080 IIF images)—1498 positive images and 582 negative images. In terms of performance, the system showed a sensitivity in the recognition of positive images equal to 92.9%, while with regard to the ability to identify the negatives, the system showed a specificity of 70.5%. Sensitivity and specificity were analyzed by varying the threshold value, in this way the ROC (Receiver Operating Characteristic) curve shown in Figure 9 was obtained. The area under the curve value obtained was: $A_Z = 0.914 +/- 0.007$. The accuracy value obtained was: Acc = 87%.

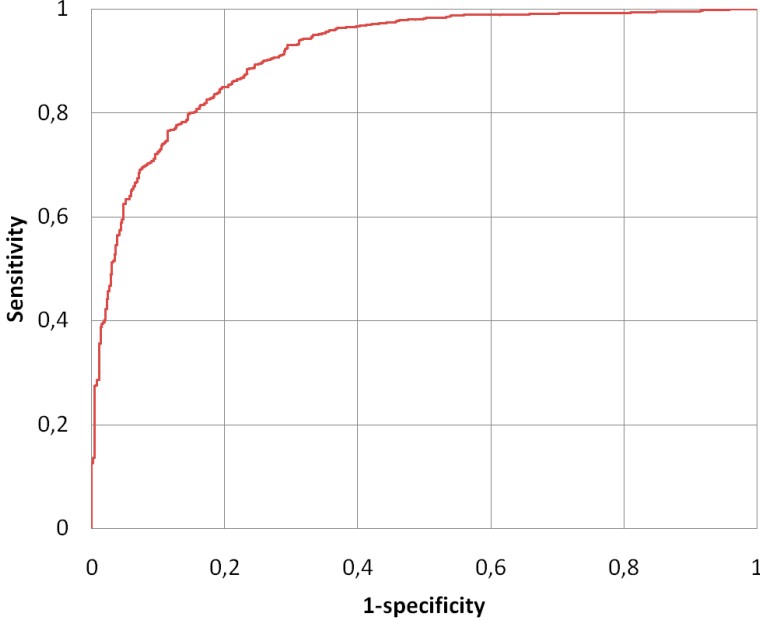

**Figure 9.** Receiver Operating Characteristic (ROC) curve of the fluorescence intensity classification method.

Since the analysis of the fluorescence intensity of the HEp-2 images has not been sufficiently addressed as a research activity, there is only a limited number of studies with which a comparison can be made. Table 4 shows the results obtained and the performance comparison with other notable intensity fluorescence classification methods, proposed in the literature, in the last years; the table reports (when present)—the number of images on which the analysis was performed, sensitivity, specificity, accuracy, and $A_Z$.

**Table 4.** Comparative fluorescence intensity analysis.

| Method | Images Dataset | Sensitivity | Specificity | Accuracy | $A_Z$ |
|---|---|---|---|---|---|
| Di Cataldo et al. [22] | 71 | – | – | 85.7% | – |
| Benammar et al. [24] | 1006 | – | – | 85.5% | – |
| Our method | 2080 | 92.9% | 70.5% | 87.0% | 91.4% |

As the intensity method works on the full slide and not on the individual HEp-2 cells, the performances did not depend on the cell segmentation method. Table 4 shows—also due to the greater statistical weight of our result—that our method of fluorescence intensity classification is, at least, comparable with the other methods analyzed.

### 3.3. Segmentation Results

The performance of the segmentation method differentiated for each patterns is shown in Table 5.

**Table 5.** Performance of the segmentation method.

| Pattern | DICE Index |
|---|---|
| homogenous | 85.6% |
| Speckled | 86.6% |
| Centromere | 81.1% |
| Nucleolar | 74.1% |
| Nuclear dots | 79.0% |
| Nuclear membrane | 76.6% |

The average Dice index, obtained by averaging the result of Dice on all analyzed images (95 images), was equal to 81.1%.

It is easy to see that, in spite of the remarkable diversity of the patterns analyzed, the method achieved very similar segmentation results for the different patterns, demonstrating a good robustness of the proposed method. The cell segmentation method presented in this paper has been compared with other segmentation methods of HEp-2 images that have been proposed in recent years. The performance comparison, in terms of the average Dice index, is shown in Table 6.

**Table 6.** Comparative performance of segmentation methods on the IIF images.

| Method | Images Dataset | Average Dice Index |
|---|---|---|
| Cheng et al. [23] | 196 | 88.9% |
| Tonti et al. [58] | 28 | 62.1% |
| Roy et al. [59] | 22 | 86.8% |
| Percannella et al. [60] | 28 | 56.8% |
| Our method | 95 | 81.1% |

It should be noted that it was possible to make a comparison with those studies that presented the Dice index as a figure of merit, or for which it was possible, from the data presented, to obtain the aforementioned index.

All of the work presented in Table 6 investigated six patterns, but the patterns typology was almost never identical; obviously, from this, derives a different segmentation-complexity. In our work, considering that the "speckled" pattern can present itself in two forms, "coarse" and "fine", conceptually the analyzed patterns, therefore, have been seven.

### 3.4. Pattern Classification Results

The performance of the pattern classification method was obtained at an image-level (i.e., in terms of the IIF images correctly classified). Table 7 shows the confusion matrix obtained for the 220 images analyzed. The classification results obtained in our experiments have been summarized and organized by the staining pattern class in Table 8.

**Table 7.** The confusion matrix for our method.

|  |  | Predicted Class | | | | | | |
|---|---|---|---|---|---|---|---|---|
|  |  | HO | SP | CE | NU | DOT | ME | TOT |
|  | HO | 19 | 0 | 0 | 2 | 0 | 0 | 21 |
|  | SP | 3 | 38 | 0 | 0 | 0 | 1 | 42 |
|  | CE | 1 | 0 | 25 | 0 | 0 | 0 | 26 |
| Actual Class | NU | 4 | 10 | 8 | 34 | 4 | 2 | 62 |
|  | DOT | 0 | 6 | 2 | 5 | 30 | 3 | 46 |
|  | ME | 3 | 9 | 1 | 0 | 0 | 10 | 23 |

**Table 8.** Per-class accuracy.

| Class | Accuracy |
|---|---|
| homogenous | 90.5% |
| speckled | 90.5% |
| centromere | 96.2% |
| nucleolar | 54.8% |
| nuclear dots | 65.2% |
| nuclear membrane | 43.5% |

The Mean Class Accuracy (MCA) obtained was equal to 73.1%, with a maximum per-class accuracy of 96.2%, for the centromere patterns, and a minimum per-class accuracy of 43.5%, for the nuclear membrane. The accuracy obtained was equal to 70.9%. Fluorescent pattern performance was evaluated not only at the image-level but also at the well-level (in terms of wells correctly classified), obtaining, in this case, an increase in performance, in particular: MCA = 75.6% and Accuracy = 73.6%.

Table 9 shows the values of accuracy, MCA, and the number of images on which the analysis was carried out, for the various CAD systems proposed in recent literature. In order to have an effective performance comparison, a further analysis was carried out, training and testing (with the leave-on-specimen-out method) on the I3Asel database originates from the First Workshop on Pattern Recognition Techniques for IIF images (I3A) [15]. The result obtained is shown in Table 9.

**Table 9.** Comparison of performance between the adopted method and previous investigations.

| Method | Images Dataset | Accuracy | MCA |
|---|---|---|---|
| Di Cataldo et al. [22] | 71 (I3A) | 73.4% | – |
| Benammar et al. [24] | 1006 | 79.3% | 79.4% |
| Cheng et al. [23] | 17 | 99.7% |  |
| Our method | 71 (I3A) | 75,1% | 78,6% |
| Our method | 220 (AIDA) | 70.9% | 73.1% |

The overall result of accuracy, obtained by analyzing the AIDA database, was negatively affected by the performance obtained on the nuclear membrane pattern; this type of pattern was not analyzed in any of the studies that were used for comparison. If the pattern nuclear membrane is not considered, the obtained performances at the image-level would be: MCA = 79.4% and Accuracy = 74.1%.

The performances obtained on the I3A database, can easily be compared with the method proposed by Di Cataldo [22], using the same database.

It should be noted that, when the analysis was conducted on a few images (for example in the work of Cheng et al. [23]), and especially if in the development the authors tune the method on the few data available to them, it is possible obtain overestimated performance values.

It is clear that in a CAD system, the classification phase is dependent on the segmentation phase; as a primary effect, the segmentation inaccuracies have differences in the values of the characteristics, and consequently, errors in the classification phase. Since the classification has not been obtained from manually segmented regions, but instead, the system is totally automatic, this consequently leads to a significant decrease in the classification performance.

## 4. Conclusions

In recent years, the need to automate the analysis of IIF HEp-2 specimens has been established, in order to obtain a fundamental tool for the diagnosis of autoimmune diseases, and to avoid the subjectivity of human interpretation. For this purpose, in this document a complete and fully automatic CAD system has been proposed, which is able to classify the IIF images, in terms of fluorescent intensity and fluorescent pattern.

To this end, our system combines the following steps:

(1) Fluorescence intensity classification: This phase performs a categorization of the fluorescent intensity into positive/negative classes. The goal has been achieved by performing a preprocessing phase of the image, extracting a considerable number of features and implementing an SVM classifier. To achieve a reduction in complexity and an appropriate selection of features, the LDA method was used.

(2) Cell segmentation: This phase of the system decomposes the input image, looking for the cells contained in it, without any a-priori knowledge about its intensity level or pattern. In order to address this problem, a method consisting of three steps has been developed—pre-segmentation, initialization by means of a randomized Hough transform, and an active contours model. The average Dice index obtained on the ninety-five images analyzed, was equal to 81.1%. In spite of the remarkable diversity of the patterns analyzed, the method achieves very similar segmentation results for the different patterns, demonstrating a good robustness of the proposed method.

(3) Pattern Classification: This phase receives the individual HEp-2 cells from the cell segmentation phase and categorizes them into a set of fluorescent patterns. For this purpose, regarding the staining patterns classification, the CAD system presented here, adopts a differentiated analysis method for each pattern. Starting from a set of preprocessing functions, all possible couplings, in terms of class accuracy, were evaluated, and the best performing combination for each pattern was chosen. For each class under analysis, a large number of features was extracted and a feature reduction phase, based on linear discriminant analysis (LDA), was performed with the aim of selecting the characteristics able to characterize the specific staining pattern. A classification approach based on the one-against-all (OAA) scheme has been implemented; six Support Vector Machine (SVM) classifiers has been implemented to classify the IIF images. The final decision-making process for the cell-staining pattern association is achieved by using a KNN classifier, having six inputs (the six outputs of SVM classifiers). The image classification was obtained by evaluating the pattern rates at the cell-level.

The developed system was evaluated on the AIDA public database, consisting of 2080IIF images, obtaining an overall accuracy of fluorescent intensity and pattern classification, respectively, around 87% and 71%. It has also been evaluated on the public database I3A, obtaining a 75.1% of fluorescent

intensity accuracy and 78.6% of pattern classification accuracy. All results were evaluated by comparing them with some of the most representative state-of-the-art work.

Unlike most of the other work in the recent literature, our system automatically analyzes all of the main steps of the ANA image analysis. The results obtained from the public database demonstrate that the system can characterize the HEp-2 specimen, in terms of intensity and fluorescent patterns, with better accuracy or is comparable with the most advanced techniques, and is proposed as a valid solution to the problem of ANA test automation.

Regarding the future developments of the system, since some patterns have demonstrated a greater difficulty in segmentation (such as nucleolar, nuclear dots, and nuclear membranes), and since it is known that the classification phase is affected by the previous phases, specific processes will be implemented to make improvements. In particular, because a correct classification of an image does not require a correct classification of all cells present, but only a correct predominance of the same, the possibility of discarding the cells (for which segmentation is doubtful), for the subsequent phases, will be evaluated, according to the appropriate characteristics. Therefore, a performance improvement can be achieved by implementing a further classifier that evaluates the goodness of segmentation of the single ROI and whether or not it is accepted for the subsequent phases.

**Author Contributions:** D.C. conceived of the study, performed the statistical analysis, and drafted the manuscript. V.T. developed the software and optimized the parameters. G.R. participated in the design, coordination, and drafting of the manuscript.

**Funding:** This research received no external funding.

**Conflicts of Interest:** The authors declare no conflict of interest.

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
