# Peer review of "An Automatic HEp-2 Specimen Analysis System Based on an Active Contours Model and an SVM Classification"

_applsci, doi:10.3390/app9020307_

Round 1
Reviewer 1 Report
In this study, the authors developed a machine learning based method to identify the cell-pattern association in Hep2-image. This paper is well-written, which makes it easy to understand, although there are some grammar and typographical issues. However, a major revision is needed to address the following concerns.
Major comments
I suggest the authors provide their source code in the supplementary information, which will be helpful for someone working in this area.
It is not clear how they construct the prediction model or what kind of cross-validation the authors have used?
Although the authors have used LDA for feature selection, but forgot to mention a few important approaches commonly used for feature selection such as sequential forward search (PMID: 30425802, 29868903), random forest algorithm (29893128, 28419290), and ANOVA feature selection (30103458). I would recommend the authors should at least mention the above approaches in the manuscript.
In figure 7, why did the authors specifically used k-NN? Did the authors use SVM for the final prediction? Anyway, clear explanation is needed.
The C values in Table 3 should be a real number rather than an integer. Fix it.
SVM has been widely used in biomedical research (PMID: 30239627, 29100375, 30081234, 30108593, 29868903, 29416743). Mention this point on page 7.
The comparative results shown in Table 9 is unfair because the size of the dataset varied significantly. I suggest the authors to use an external dataset to validate their predicted model and compare with the existing methods if the method is publicly available.
Section 2.14 is not needed because you are not providing any web server.
Minor revision.
ANA is not defined in the introduction part.
In pg 3, line 117 and 118 respectively contains 85,5% and 79,3%. Double check the values throughout the manuscript.
Acronyms should be used wisely. Pg 3, line 124: Support Vector Machine classifier (SVM). This has been repeated in pg 4and 6.
Author Response
Reviewer 1
In this study, the authors developed a machine learning based method to identify the cell-pattern association in Hep2-image. This paper is well-written, which makes it easy to understand, although there are some grammar and typographical issues. However, a major revision is needed to address the following concerns.
Major comments
I suggest the authors provide their source code in the supplementary information, which will be helpful for someone working in this area.
I thank the Reviewer for the suggestion. However, since the developed system is composed of many parts (analysis and optimization of fluorescence intensity, segmentation, features extraction and reduction, classifications, optimizations and tests) not all written with the same programming language (it was mainly used c ++ and matlab) at present unfortunately the code would prove impossible to use and consequently of poor help.
It is not clear how they construct the prediction model or what kind of cross-validation the authors have used?
In order not to introduce bias in the analysis, the public database part was used only for the test (also to allow future comparisons with other methods). The private database part was used for training-tuning. In this last phase, in order to make the best use of the data, the leave-one-specimen-out cross validation technique have been used. The method consists in leaving out one specimen, rather than leaving out a single image (or a single cell) for the construction of the training set; images of the same specimen, belonging to the same patient, are similar (in terms of the average intensity and contrast) and introduce bias. The specimen left out is used for validation.
Regarding the analysis of fluorescence intensity, as in this case the wells at our disposal were statistically much greater, during training-tuning we proceeded with a leave-ten-specimens-out (i.e. leaving out ten specimen).
Clarification added in the paper.
Although the authors have used LDA for feature selection, but forgot to mention a few important approaches commonly used for feature selection such as sequential forward search (PMID: 30425802, 29868903), random forest algorithm (29893128, 28419290), and ANOVA feature selection (30103458). I would recommend the authors should at least mention the above approaches in the manuscript.
Thank you for your suggestion. Further clarifications regarding the features selection have been added to the paper and the recommended references have been included in the bibliography
In figure 7, why did the authors specifically used k-NN? Did the authors use SVM for the final prediction? Anyway, clear explanation is needed.
In multiclass classification problems (specifically with OAA scheme) the outputs of the binary n-classifiers are evaluated; usually the classifier that produced the maximum output value is identified, and the final association of the generic ROI is assigned to the relative class. This procedure may not be very robust, for example it could happen that one of the classifiers produces output values on average higher than the other classifiers, thus completing the final classification. In this work it was decided to evaluate the outputs produced by means of a further classifier. The classifier KNN has been chosen as it allows a simple multi-class implementation; this classifier, using examples belonging to the classes to be analyzed, associates the generic element with the class having the most examples close to it.
The clarification was added to the paper.
The C values in Table 3 should be a real number rather than an integer. Fix it.
Done.
SVM has been widely used in biomedical research (PMID: 30239627, 29100375, 30081234, 30108593, 29868903, 29416743). Mention this point on page 7.
Done and references have been added to the bibliography.
The comparative results shown in Table 9 is unfair because the size of the dataset varied significantly. I suggest the authors to use an external dataset to validate their predicted model and compare with the existing methods if the method is publicly available.
As suggested, in order to have an effective performance comparison, a further analysis was carried out, training and testing (with the leave-on-specimen-out method) on the I3Asel database dataset originates from the First Workshop on Pattern Recognition Techniques for IIF images (I3A) [15]. The result obtained was added in table 9.
Section 2.14 is not needed because you are not providing any web server.
The section has been removed as suggested.
Minor revision.
ANA is not defined in the introduction part.
Done.
In pg 3, line 117 and 118 respectively contains 85,5% and 79,3%. Double check the values throughout the manuscript.
Check done
Acronyms should be used wisely. Pg 3, line 124: Support Vector Machine classifier (SVM). This has been repeated in pg 4and 6.
Repeats removed.
Reviewer 2 Report
This paper describes an observer independent method to characterize IIF images based on image processing and Machine learning method, then tested with a set of IIF images and finally compared with other works. Its a good work however I have the following concern on the paper before final decision. First of all, it is has 29% similarity index according to iThenticate. Pls revise the draft thoroughly. Q1. L423, "The features an be" should be "The features can be" Q2. pls mention some result of the extracted features. Q3. Section 2.7, How dimensionality reduction was performed. and why it was needed, particularly in this work. please state clearly. Q4. ".....27 characteristics obtained at 4 different quantization levels, for a total of 108 features." Pls mention clearly the feature's name those are used in this work. Q5. Principal component analysis is popular for dimensionality reduction, Why LDA was used here.Author Response
Reviewer 2
This paper describes an observer independent method to characterize IIF images based on image processing and Machine learning method, then tested with a set of IIF images and finally compared with other works. Its a good work however I have the following concern on the paper before final decision.
First of all, it is has 29% similarity index according to iThenticate. Pls revise the draft thoroughly.
Done.
Q1. L423, "The features an be" should be "The features can be"
Thanks, correct.
Q2. pls mention some result of the extracted features.
A well-structured features extraction process helps to achieve good results in the classification phase; the results presented in paragraph 3 are also due to the extraction and selection of features.
Q3. Section 2.7, How dimensionality reduction was performed. and why it was needed, particularly in this work. please state clearly.
It is known that if in a supervised process the number of inputs of a classifier is doubled not increasing the classifying power of the same (for example considering each features twice) the classification process will be less performing because more difficult in the training phase to find hypersurfaces separation; in the case of SVM classifiers, it will be more difficult to find the optimal parameters of the classifier. In general, whenever the information of a single feature is completely contained in the others, there is an increase in classification complexity and therefore the above is valid. The LDA is based on this approach. Conceptually, for the i-th features, the discriminating power of the n features and n-1 features (obtained by removing the i-th featues) is evaluated with the Fisher criterion. In the case in which the first set does not have a discriminating power greater than the second, dimensionality is reduced by eliminating the relative features. The same operation is carried out for all the features.
Q4. ".....27 characteristics obtained at 4 different quantization levels, for a total of 108 features." Pls mention clearly the feature's name those are used in this work.
They are reported at the end of section 2.7 Features extraction
Q5. Principal component analysis is popular for dimensionality reduction, Why LDA was used here.
PCA is used when dealing with a single class of elements, for example for unsupervised problems (PCA ignores class labels). LDA is a supervised.
https://sebastianraschka.com/faq/docs/lda-vs-pca.html
Round 2
Reviewer 2 Report
I have the following minor comments. Please revise the draft accordingly.
[1] For better understand for reader, please mention reason of using 4 different quantization levels clearly in the context.
[2] The classification accuracy of the method deteriorates due to poor segmentation, specifically, for the last three patterns (NU, DOT, ME). Please mention the challenges behind it and future task how to improve it.
[3] Table 9, Include reference for 4th method (using 13A dataset)
[4] L840, which is correct- patter or pattern?
[5] Figure 1, please insert sub-caption for every figure, i.e. (a), (b), ..
[6] page 7, L 446 & 447, clearly state the function or library used for this work.
[7] Fig 4, (C) should be (c)
[8] L 530, there should be a space "identified by"
[9] Table 2, the name function 1 and function 2 is confusing, Please clarify them in the context.
[10] Fig 8, There is decision box in the last which is incomplete.
[11] Remove the title (ROC curve) in the figure 9.
[12] Please prove the possible reason/ challenges for lower segmentation result for NU, DOT and NM.
[13] There are also some successful implementation of SVM and KNN based CAD system in medical applications (https://doi.org/10.1016/S1474-422(13)702631; 10.20965/jaciii.2018.p0249); which shows a good reason for potential use of this machine learning method for cell classification.
[14] Finally concise the conclusion or it can marge with the result and discussion section. Then rewrite it.
Author Response
The documeto has been modified as suggested.
We want to thank the Reviewer for his attention in reading the document and for the numerous suggestions thanks to which the document has greatly improved.